

# Enhanced neural network classification for Arctic summer sea ice

Anne Braakmann-Folgmann[1], Jack C. Landy[1], Geoffrey Dawson[2], and Robert Ricker[3]

[1]UiT The Arctic University of Norway, 9019 Tromsø, Norway
[2]University of Bristol, School of Geographical Sciences, Bristol BS8 1SS, UK
[3]NORCE Norwegian Research Centre, 9019 Tromsø, Norway

**Correspondence:** Anne Braakmann-Folgmann (anne.bf@uit.no)

**Abstract.** Lead/floe discrimination is essential for calculating sea ice freeboard and thickness (SIT) from radar altimetry. During the summer months (May-September) the classification is complicated by the presence of melt ponds. In this study, we develop a neural network to classify CryoSat-2 measurements during the summer months, building on the work by Dawson et al. (2022) with various improvements: (i) we expand the training dataset and make it more geographically and seasonally diverse, (ii) we introduce an additional thinned floe class, (iii) we design a deeper neural network and train it longer and (iv) we update the input parameters to data from the latest publicly available CryoSat-2 processing baseline. We show that both the expansion of the training data and the novel architecture increase the classification accuracy. The overall test accuracy improves from $77 \pm 5$ % to $84 \pm 2$ % and the lead user accuracy increases from $82 \pm 10$ % to $88 \pm 5$ % with the novel classifier. When used for SIT calculation, we observe minor improvements in agreement with the validation data. However, as more leads are detected with the new approach, we achieve better coverage especially in the marginal ice zone. The novel classifier presented here is used for the Summer Sea Ice CryoTEMPO (CryoSat Thematic Product).

## 1   Introduction

Sea ice thickness (SIT) determines the overall sea ice volume, the stability of the sea ice, biological growth under the ice and is an essential variable for shipping and the safety of marine infrastructure. Estimates of SIT during the summer months improve forecasts of sea ice concentration and extent (Zhang et al., 2023). Summer is also the season when most ships cross the Arctic and when most biological production takes place (Arrigo et al.). This makes estimates of SIT during the summer particularly valuable. However, estimating SIT from satellite altimetry during summer is significantly more challenging than it is in winter due to melt ponds, which develop from melting snow and sea ice. One major challenge is to distinguish between waveform returns from melt ponds and leads, as both generate a specular reflection (Drinkwater, 1991). If melt ponds are misclassified as leads, freeboard can be biased low as the melt ponds are usually located above sea level. Furthermore, the highly reflective melt pond surface dominates the return signal – even if melt ponds cover only a small fraction of the footprint area (Landy et al., 2022). As the melt ponds are typically located below the average floe height, this introduces another negative bias, which is also referred to as electromagnectic (EM) bias. It is largest for rougher sea ice, where the difference in elevation between the melt pond and the average floe height is highest, and for lower melt pond fractions (Landy et al., 2022; Dawson and Landy, 2023). Another challenge in summer is the lack of an operational snow depth product covering the summer months (Landy



et al., 2022; Zhou et al., 2021). Owing to these additional challenges, beyond conventional "winter" sea ice processing, the standard processing must be adapted for summer months and research on summer SIT is still developing.

Radar altimetry measurements are generally classified as lead or floe by exploiting the different waveform shapes (Drinkwater, 1991; Laxon, 1994). In the absence of melt ponds, only leads produce a strongly specular nadir reflection, yielding a peaky
waveform, while snow-covered sea ice floes produce more diffuse reflections yielding a wider waveform shape (Laxon, 1994). Open ocean returns are also wider, but these can be discarded by applying for example a sea ice concentration mask (Paul et al., 2018; Poisson et al., 2018; Tilling et al., 2018) - sometimes together with additional waveform thresholds (Ricker et al., 2014). For the remaining waveforms in sea ice covered waters, empirical thresholding techniques have been commonly applied to discriminate between leads and floes. The thresholds are typically chosen conservatively to leave some margin for ambiguous
waveforms; however, these ambiguous waveforms can inadvertently filter out certain ice types, like thin sea ice (Müller et al., 2023). Pulse peakiness is the most frequently used parameter for this discrimination (Laxon et al., 2013; Ricker et al., 2014; Armitage and Ridout, 2015; Paul et al., 2018; Poisson et al., 2018; Tilling et al., 2018; Swiggs et al., 2024; Zygmuntowska et al., 2013; Lee et al., 2016). Additional waveform parameters include backscattered power ($\sigma_0$) (Zygmuntowska et al., 2013; Paul et al., 2018; Poisson et al., 2018; Lee et al., 2016), stack standard deviation (Laxon et al., 2013; Ricker et al., 2014; Tilling
et al., 2018; Swiggs et al., 2024; Lee et al., 2016), stack kurtosis (Ricker et al., 2014; Lee et al., 2016), leading edge width (Armitage and Ridout, 2015; Paul et al., 2018) and trailing edge width (Zygmuntowska et al., 2013). Passaro et al. (2018) suggested stack peakiness as a single parameter, yielding comparable performance to the multi-paramter approach by Ricker et al. (2014). However, stack peakiness is not routinely available in the ESA L1B product for CryoSat-2 Synthetic Aperture Radar (SAR) and SAR Interferometric (SARIn) modes (it was produced by the SAR Versatile Altimetric Toolkit for Ocean
Research and Exploitation (SARvatore) processor, which is no longer freely available). Röhrs et al. (2012) use only the maximum power of the waveform to detect leads, which Wernecke and Kaleschke (2015) found to give superior results to other waveform parameters. All these threshold based techniques (including decision trees), however, rely on tuning the parameters for the specific conditions where they are applied (Dettmering et al., 2018; Lee et al., 2018).

Beyond simple thresholding techniques, Lee et al. (2018) suggest employing spectral mixture analysis to CryoSat-2 wave-
forms, i.e. describing each waveform as a mix of lead and floe endmembers. This method circumvents the need to readjust thresholds for different baselines, seasons or regions, but has only been applied to winter and shows similar performance to thresholding methods. Müller et al. (2017) applied an unsupervised clustering approach based on waveform maximum, width, noise, leading edge slope, trailing edge slope and trailing edge decline. The clusters are then manually labelled as leads or floes and a K-nearest neighbor approach is used to classify new samples. Although originally designed for Envisat and SARAL data,
this approach has also been applied to CryoSat-2 SAR data (Dettmering et al., 2018). And finally, Poisson et al. (2018) trained a neural network to find leads in ENVISAT data using seven parameters, which fully describe the waveform. Compared to a multi-criteria approach, they found the neural network to discard fewer samples as ambiguous.

During the summer months, the classification is complicated by the fact that both melt ponds and leads generate a specular reflection Kwok et al. (2018). Therefore, simple thresholding techniques are not applicable to the melt season. A convolutional
neural network (CNN) developed by Dawson et al. (2022) was the first and so far only attempt for a summer specific clas-



sification achieving an accuracy of 80%. This method differed from previous approaches by calculating normalized profiles of waveform parameters over short 3-km (11-sample) windows and delivering sets of these parameters to the CNN. We are building on this study, but present several improvements further boosting the classification accuracy and mitigating the effect of melt ponds. Firstly and most importantly, the number of training samples is multiplied and made more geographically and

seasonally diverse. Secondly, an additional thinned floe class with distinct characteristics is added. Third, the new CNN is trained with publicly available ESA data rather than the discontinued SARvatore processed data and using the latest CryoSat-2 baseline E. Finally, a deeper convolutional neural network (CNN) is built – making use of the additional training data and allowing for a better representation (Schmidhuber, 2015) - and the training process is extended to allow for additional fine-tuning of the parameters. We demonstrate that these changes improve the classification accuracy and limit the number of melt ponds

that are misclassified as leads. SIT estimates using the novel classifier with the methods from Landy et al. (2022) are validated and compared with SIT estimates using the original classifier (Dawson et al., 2022).

## 2 Data

### 2.1 CryoSat-2

CryoSat-2 is a radar altimeter launched by the European Space Agency (ESA), which has been measuring the two-way travel

time between the satellite and the Earth's surface since October 2010. Applying range corrections and retracking the echo waveform yields surface elevations with respect to an Earth ellipsoid. Over sea ice, the radar freeboard can be derived from the relative elevation difference between sea ice floes and leads. We use CryoSat-2 baseline E data available from the ESA Science Server. Both SAR and SARIn mode observations are used. The main input to our processing chain are the Level 1B files, however, we also use some variables from the corresponding Level 2 files (e.g. pulse peakiness and geophysical corrections).

Data between 1 May and 30 September is regarded as the summer season. Our processing is applicable to all months, but is validated and optimized specifically for the summer season. In terms of spatial coverage, we include all data from the Arctic Ocean above 45° North and where the sea ice concentration (taken from the Level 2 files and originally sourced from OSI-SAF) exceeds 30%.

### 2.2 Optical and radar imagery

We identify near-coincident optical and radar satellite images from Sentinel-1, Sentinel-2, Landsat-8 and RADARSAT-2 to build a database of known lead and floe samples. These images spatially overlap with CryoSat-2 passes, and all images captured within one hour of a CryoSat-2 overpass are selected and manually evaluated. For every paired satellite image and CryoSat-2 track we obtain the general sea ice drift conditions for the relevant location from Polar Pathfinder Daily 25 km EASE-Grid Sea Ice Motion grids, Tschudi et al. (2020). For all pairs with time difference >15 minutes, we only retain the data in low sea ice

drift conditions when the drift was < 0.3 m/s.



For the optical data (Landsat-8, Sentinel-2A and -2B) we use true colour images and for the SAR images (RADARSAT-2, Sentinel 1A and 1B) we analyse both the HH and HV channels. The Sentinel-1 images are pre-processed by employing border and thermal noise reduction, speckle filtering with the refined Lee filter and are converted to sigma naught ($\sigma^0$ ,Filipponi,2018). The RADARSAT-2 images are also calibrated to $\sigma^0$ and filtered with a Lee sigma filter. While originally, a similar number of

images were found across all summer months, many images had to be discarded (e.g. because of cloud cover, covering only open water, too low contrast, small overlaps or poor agreement with the CryoSat-2 data, see Methods). Overall, 298 Sentinel-1 images, 4 RADARSAT-2 scenes and 3 Landsat-8 images are used to create the dataset presented in Dawson et al. (2022). As part of the training database expansion, additional Sentinel-1 and Landsat-8 images are added, yielding a total of 1011 usable Sentinel 1 images, 4 RADARSAT-2 scenes and 38 Landsat-8 images. All samples from Sentinel-2 had to be discarded. Many

of the images yield multiple training/testing samples, so the number of samples is significantly higher than the number of images.

## 2.3    Validation and comparison data

Independent validation data is essential to assess the performance of our summer freeboard calculation. Here, we combine airborne surveys and mooring data from various different regions covering different sea ice types and all summer months

(May to September, included). We also compare to results to the Pan-Arctic Ice Ocean Modeling and Assimilation System (PIOMAS) for an independent assessment of the resulting overall sea ice volume.

### 2.3.1    Airborne data

We use airborne measurements of sea ice freeboard from NASA's Operation Ice Bridge (OIB) campaign on 16 and 19 July 2016 and 24 and 25 July 2017 Studinger (2014). The OIB laser scanner freeboards are derived from raw height observations using

the method described in Dawson et al. (2022) (Supplementary material 1). The final freeboards are averaged from all valid observations along 7-km sections of the flight track, to mirror the length scale used in obtaining a single freeboard estimate for our summer CryoSat-2 processor.

We also use direct SIT measurements from the Alfred-Wegener-Institute (AWI)'s airborne ICEBird surveys, acquired using electromagnetic induction sounding (AEM, Belter et al. (2021)). The considered flights were conducted between 24 July and

1 August 2016, 13 and 28 August 2017, and 31 July and 13 August 2018. The OIB laser scanner freeboards are derived from raw height observations using the method described in Dawson et al. (2022) (Supplementary material 1). The final freeboards are averaged from all valid observations along 7-km sections of the flight track, to mirror the length scale used in obtaining a single freeboard estimate for our summer CryoSat-2 processor.

The main processing steps for the AEM data are described in (Belter et al., 2021) and the only further step applied here, as

in Landy et al. (2022), is to remove airborne observations where the range to the surface is >30 m or the observation is flagged in the raw data file.

All airborne data are sampled to an 80-km resolution grid at biweekly (twice monthly) time intervals matching the gridded CryoSat-2 observations (see below). A simple binned mean is used for gridding. Comparisons with the CryoSat-2 data are



**Figure 1.** Distribution of samples used in the training data base by (Dawson et al., 2022), subplots a, b) and the extended training database (c, d) divided by month (a, c) and class (b, d)





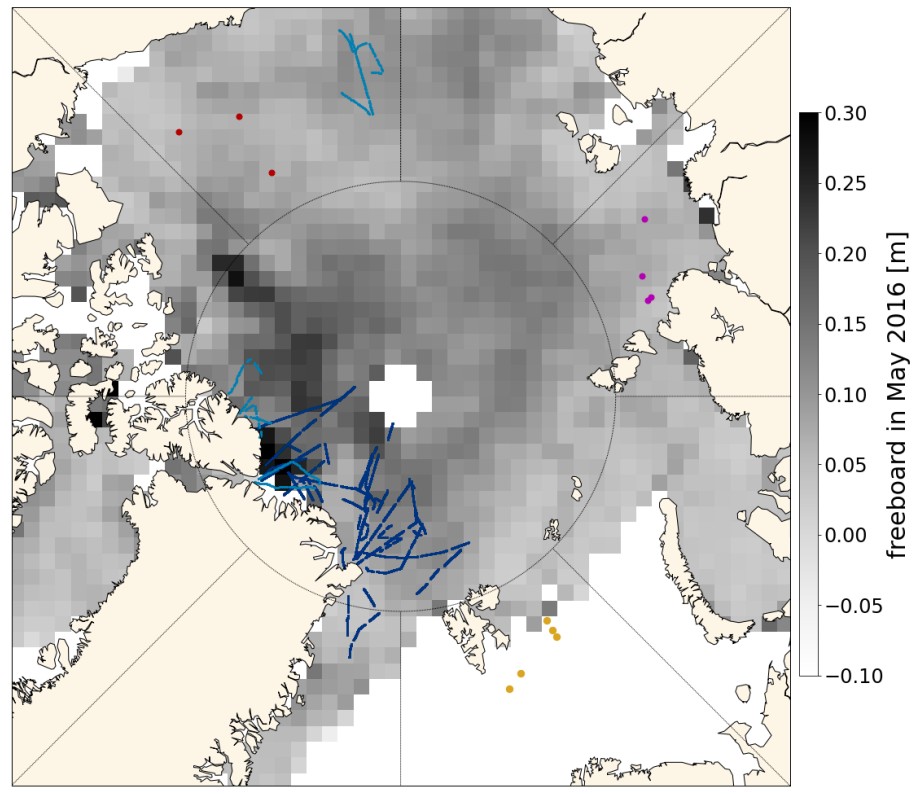

**Figure 2.** Validation data used for comparison: OIB flights from 2016 and 2017 in turquoise, EM-bird flights from 2016-2018 in blue, BGEP moorings data from 2015-2018 in red, Laptev Sea moorings from 2015-2016 in pink and Barents Sea moorings from 2015-2019 in yellow

made per biweekly grid cell that includes both valid satellite and airborne observations. This is identical to the method used in
Landy et al. (2022).

### 2.3.2 Mooring data

We use the measurements from three upward looking sonars (ULS) of the Beaufort Gyre Exploration Program (BGEP) in the
Beaufort Sea (https://www2.whoi.edu/site/beaufortgyre/), ULS and acoustic Doppler current profiler (ADCP) data from four
moorings in the Laptev Sea deployed by the Alfred-Wegener-Institute (Belter et al., 2019) and ULS data from five moorings in
the Barents Sea deployed by Equinor (Onarheim et al., 2024) (Figure 2). Two of the Laptev Sea moorings cover 2015 and 2016
and another one covers 2015 only. From the Beaufort Sea and Barents Sea Moorings we use data from five years (2015-2019),
but depending on the year, the Barents Sea is only sea-ice covered until May-July. As the airborne data cover July-August,



this makes a valuable addition to the overall dataset, though, and balances the temporal coverage. We use the daily mean observations provided in all raw datasets and discard measurements of zero draft (open ocean). Biweekly ice drafts are then calculated from the daily observations on a matching timescale to the gridded CryoSat-2 observations. All valid CryoSat-2 grid cells within 150 km of each mooring, at each biweekly timestep, are averaged to compare to the concurrent mooring time series. This search radius is selected to attempt to represent sea ice conditions for all floes drifting over the mooring within a two week interval.

### 2.3.3 PIOMAS

Reanalysed model estimates of sea ice volume were obtained from the Pan-Arctic Ice Ocean Modeling and Assimilation System (PIOMAS). PIOMAS assimilates sea ice concentration and sea-surface-temperature data to produce an optimized model reanalysis of the Arctic sea ice cover based on reanalysed atmopsheric forcing. Daily PIOMAS effective sea ice thickness and sea ice concentration grids are available from the Applied Physics Laboratory Version 2.1 reprocessed model fields at http://psc.apl.uw.edu/research/projects/arctic-sea-ice-volume-anomaly/data/model_grid. Here, we calculate pan-Arctic PIOMAS sea ice volume on biweekly time intervals, covering a common grid to the valid CryoSat-2 observations, for a fair comparison. This is identical to the method used in Landy et al. (2022).

### 2.3.4 SnowModel-LG

Estimates of snow depth and snow density are needed to compare the radar freeboard estimates from CryoSat-2 to measurements of sea ice freeboard, draft, thickness or volume from the in-situ and model data. There is a general lack of operational snow depth (and density) estimates during the summer months, as both passive microwave estimates and dual frequency altimetry estimates are limited to the winter season (Zhou et al., 2021; Markus et al., 2006; Garnier et al., 2021). Here, we use data from SnowModel-LG available from NSIDC (Liston et al., 2020). The daily SnowModel-LG data are averaged to biweekly intervals and resampled to the same 80-km grid as the CryoSat-2 data for the freeboard-to-ice thickness/draft conversion (see below).

## 3 Methods

The neural network developed by Dawson et al. (2022) forms the basis for our new CNN. Compared to the original approach, we increase the volume of training data, we add an additional class for thinned floes, and we design a deeper neural network architecture including a longer training process. Furthermore, the training data is updated to use the latest Baseline E data from CryoSat-2, which is publicly available, making it more current and reproducible. In this section we describe both the original configuration and the improvements made in this study, but focus on the latter. For more information on the original configuration, the reader is referred to Dawson et al. (2022).



## 3.1 Generation of the training database

A database of known lead and floe samples is created using manual classification in coincident optical and radar satellite imagery (see Dawson et al. (2022)). While Dawson et al. (2022) used 500 unique samples of known leads, 'good' floes and 'noisy' floes (120, 218 and 162 respectively), we increase the number of known samples more than fivefold to 2622 by adding another year (2021), an additional class for thinned floes and by increasing the time limit between the coincident optical and SAR images from 15 to 60 minutes. As sea ice drift speeds are typically well below 0.2 m/s in the Arctic (Kwok et al., 2013), a large portion of the data will not be significantly mis-aligned (less than 720 m offset). Nevertheless, we use ice motion data (from Polar Pathfinder Daily 25 km EASE-Grid Sea Ice Motion grids, Tschudi et al. (2020)) to approximate drift distance and do not include samples where the drift speed is greater than 0.3 m/s (i.e., >1000 m in an hour). The alignment between images and tracks is further verified manually while training samples are selected, and mis-aligned tracks are discarded. The new extended database contains 457 leads, 284 thinned floe samples, 1187 good floes and 694 noisy floes. The manual selection of leads and floes in the images follows the same procedure as described in Dawson et al. (2022). In the optical data leads are identified based on their intensity and shape. Compared to melt ponds they are usually elongated, whereas melt ponds tend to be round. Furthermore, the difference in reflectance of the red and blue channels is analysed, since melt ponds reflect more blue light compared to leads (Istomina et al., 2016). In the HH channel of SAR data, leads can appear either bright or dark, depending on the incidence angle and wind speed. In the HV image, in contrast, leads almost always appear dark, but the contrast to the surrounding ice is lower. Leads are picked in the SAR images where they can be discriminated clearly in either channel.

All manual picks are then compared with the along-track CryoSat-2 data and, for leads, only those picks that exhibit a notable change in elevation are retained. The manual picks of floes are then separated into 'thinned floes', 'good floes' and 'noisy floes' according to the CryoSat-2 data. Samples where a decrease in elevation is observed (as for leads), but the backscatter and additional waveform parameters show the opposite behaviour to lead samples, are labelled as 'thinned floes' and characterised as thinner sea ice floes surrounded by thicker floes (Figure 3). When the CryoSat-2 data exhibits relatively-large variations in parameters along track, the samples identified as floes from the imagery were classed as 'noisy floes'. We assume these variations come from true geophysical sources like sea ice surface roughness, but also mixed melt conditions with some diffusely-scattering and some pond-dominated specular floes within the along-track window. Floes with relatively-lower variation along track are just labelled as 'good floes' (Figure 3). The introduction of these additional classes ensures that the samples from each class are sufficiently similar and helps the classification, but thinned floes, good floes and noisy floes are later merged into a single ice class for freeboard calculation.

## 3.2 CNN architecture and training

As input to the CNN we use the along-track variation of five variables including elevation, backscatter and waveform parameters (see Figure 3). Eleven measurements along the track are used, centrered around the point to be classified, and the anomalies of elevation, backscatter, pulse peakiness, stack scaled amplitude and stack centre angle are calculated. Pulse peakiness is the




**Figure 3.** Sea surface height (SSH), backscatter (Sig0), pulse peakiness (PP), stack amplitude (stack amp) and stack centre angle as anomalies (rows) for 11 points along-track sampled around the manually classified point (x-axes) are used as input to the CNN. These variables exhibit unique signatures for the four different classes (columns). The grey lines are the data from the individual samples and the blue line is the average per class.





most widely-used parameter to differentiate between leads and floes in winter. Also backscatter and other stack parameters have been employed for winter. The first summer classifier by Dawson et al. (2022) used range integrated power peakiness instead of stack scaled amplitude and stack centre angle. The choice of these variables was updated to be applicable to ESA L1b and L2 data rather than SARvatore processing.

The samples from each class exhibit a unique signature in all along-track parameters (Figure 3). While both leads and thinned
floes are characterised by a lower elevation compared to the surrounding points, leads have a higher backscatter and the return waveform is peakier than the surrounding floes. The same holds for stack amplitude. In contrast, thinned floes exhibit a lower backscatter than their surrounding, lower pulse peakiness and a lower stack amplitude. From this figure, it is evident that the newly introduced thinned floe class has its own unique characteristics. However, the current CNN is designed to be sensitive to along-track parameter variations, so it cannot detect large areas of thinned floes along the track. What is classified as thinned
floe, is thinner than the surrounding rather than the absolute thinnest ice.

The neural network consists of a number of 1D-convolutional layers with kernel size 3, rectified linear unit (relu) activation function and max pooling layers of kernel size 2, before a final dense layer with softmax activation function generates the final output. Originally, Dawson et al. (2022) proposed a neural network architecture with three convolutional layers and one pooling layer steadily reducing the length of the sequence as shown in Figure 4. The output is a three by one matrix containing
210 the probability for the sample to represent a lead, a good floe or a noisy floe. The highest probability is used to classify each point as lead, good floe or noisy floe. When using the extended training data set with this architecture, we simply change the output to a four by one matrix containing the additional thinned floe class. When freeboard is calculated, samples from the thinned floe, good floe and noisy floe classes are merged into one ice class, but their initial separation aids the classification, as they exhibit different characteristics (see Figure 3).

In this study, we revisit the design of the neural network architecture. We add more layers and increase the number of feature maps in the deeper layers, to allow the neural network to learn more complex relationships. "Same" padding (repeating edge values to maintain the length of the sequence) is added to each of the convolutional and pooling layers, to keep the same length of the data except during pooling and enable a deeper architecture. On the other hand, to avoid overfitting, we also add dropout of 0.3 (Figure 5).

For the training process, we randomly split the dataset into training (80%), validation (10%) and test data (10%). The network is then trained with the Root Mean Squared Propagation (RMSProp) optimizer using a categorical cross entropy loss function at an initial learning rate of 0.001. Originally, the network was trained for 10 epochs only (Dawson et al., 2022). With the larger training database and the deeper neural network architecture, however, it is sensible to increase the training time. To enable a longer training process, while still ensuring that we do not overfit the data, we halve the learning rate, when no improvement
on the validation data has been recorded for 8 epochs and we automatically stop the training once the validation loss has not improved for 20 epochs. The latter allows for fine-tuning the parameters as the algorithm approaches a minimum of the loss function.




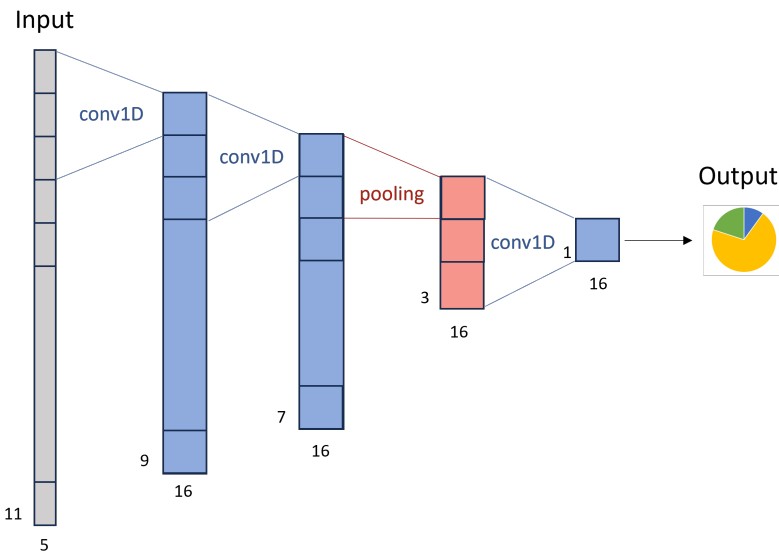

**Figure 4.** Original CNN architecture from Dawson et al. (2022)

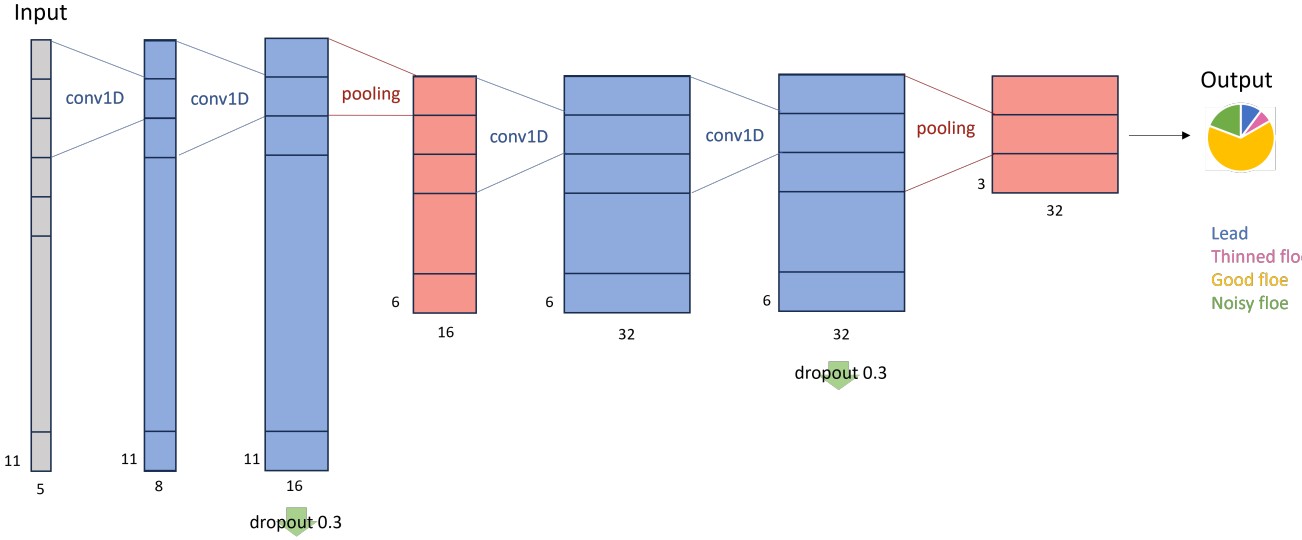

**Figure 5.** Novel CNN architecture with additional layers and feature maps, same padding, dropout and an additional output class





### 3.3 Summer sea ice freeboard and thickness calculation

The summer sea ice freeboard calculation has been described by Dawson et al. (2022) in detail and the conversion to SIT
follows the approach by Landy et al. (2022). Here, we only provide a brief summary and point out differences to Dawson et al.
(2022) and Landy et al. (2022).

First, all waveforms are retracked using the threshold first maximum retracking algorithm (TFMRA) with a 50% threshold
used for both leads and floes (Helm et al., 2014; Ricker et al., 2014). While the SARvatore data used by Dawson et al. (2022)
were retracked with SAMOSA+, using TFMRA speeds up the processing and was required to be used for the CryoTEMPO
operational product. Next, we compute surface elevations by subtracting the retracked range from the height of the satellite and
applying the following range corrections: ionosphere correction, wet and dry troposphere correction, inverse barometric cor-
rection, elastic ocean tide, long-period ocean tide, ocean loading tide, solid earth tide and geocentric polar tide. All corrections
are taken from the CryoSat-2 Level 2 files. We also subtract the mean sea surface height (again taken from the Level 2 files)
to produce a sea surface height anomaly (SSHA) and remove long-wavelength signals using a rolling median of 30 points.
Then we classify each measurement using either the former or the new CNN, as described in the previous section. As the lead
classification is occasionally offset by plus or minus one point along-track, we also label the two neighbouring samples as leads
(dilation by 1). Finally, radar freeboard is calculated at each lead location. We fit a robust 2nd order polynomial using the huber
loss function to all floe heights within a 7 km window centered around the lead points, yielding the average floe height $h_{ice}$.
Subtracting the surface elevation of the lead points $h_{lead}$ from the average height of the ice samples then yields radar freeboard
$h_{rfb}$. We only use the largest estimate of radar freeboard for each group of three or more consecutive lead points.

$$h_{rfb} = h_{ice} - h_{lead} \tag{1}$$

Unrealistic freeboard estimates are filtered out, removing estimates where the freeboard/backscatter exceeds the median +/-
4 times the median absolute deviation of all freeboard/backscatter measurements within 80 km and 16 days. We also remove
freeboard estimates below -0.1 m, above 1 m and where the root mean squared error (RMSE) of the polynomial fit was greater
than 0.25 m. We sample all valid radar freeboard observations, on a biweekly basis, to an 80-km grid on the North Polar
Stereographic projection using inverse-distance weighted averaging. All observations within 80 km of the grid cell center, i.e.,
twice the radius of the grid cell, are used to compute the cell value. Finally, we apply a correction to account for the EM bias.
The bias correction is calculated from radar waveform model simulations as a function of surface roughness and melt pond
fraction, obtained from auxiliary observations as in Landy et al. (2022).
As the airborne validation data measures either sea ice freeboard or SIT, the moorings record sea ice draft, and PIOMAS
yields estimates of sea ice volume, assumptions are required to assess their agreement with our estimates of radar freeboard.
For the radar freeboard to ice draft or thickness conversion, we apply an identical approach to Landy et al. (2022). To convert
the bias-corrected radar freeboard $h_{rfb}$ to sea ice freeboard $h_{fb}$, we apply a correction for the slower travel time of the radar
signal through the snow layer assuming a constant penetration $\delta_p$ of 90% of the snow depth $h_s$. Snow depth $h_s$ and snow
density $\rho_s$ estimates are taken from SnowModelLG (Liston et al., 2020).



$$h_{fb} = h_{rfb} + \delta_p h_s((1 + 0.51\rho_s/1000)^{1.5} - 1) \tag{2}$$

To convert freeboard to SIT, we use sea ice-type dependent ice densities $\rho_i$ of 917 and 882 kg/m3 for first-year ice and multi-year ice, respectively, and a sea water density $\rho_w$ of 1024 kg/m3. Sea ice draft is then obtained simply from the ice thickness minus the ice freeboard. Uncertainty on the thickness or draft is estimated as in Landy et al. (2022). For the comparison with

PIOMAS, the gridded CryoSat-2 sea ice thickness data are finally multipled by ice concentration from the OSI SAF 'OSI-450' climate data record (available from https://osi-saf.eumetsat.int/products/osi-45070) and the grid cell area to obtain a sea ice volume time series.

$$h_{\mathrm{SIT}} = \frac{h_s\rho_w - h_{fb}\rho_w - h_s\rho_s - \delta_p h_s\rho_w}{\rho_i - \rho_w} \tag{3}$$

$h_{draft} = h_{SIT} - h_{fb}$                                                                          (4)

Another challenge when comparing satellite data to in-situ data is the difference in spatial and temporal sampling. For the summer months the spatial resolution of the CryoSat-2 estimates is even lower than during winter. This is on the one hand because we calculate freeboard at lead locations rather than for each floe sample and on the other hand because the data is noisier and therefore requires large-scale averaging. For the comparison with airborne and in situ data, we therefore

downsample the in situ observations to approximately the scale of the gridded CryoSat-2 data, as described in Section 2 above.

## 4 Results and Discussion

### 4.1 Classification results

First, we assess the classification accuracy based on the random split of unseen test data. We use the original architecture and setup as a baseline ('D22', Dawson et al. (2022) ) and assess the impact of adding more training data ('more data') and

the deeper architecture and longer training separately ('this study'). Note that the baseline performance differs slightly from the results presented by Dawson et al. (2022) due to the slight modification of input parameters, using the ESA L1b and L2 files and a TFMRA retracker rather than SARvatore files, and the update to baseline E. Therefore, we also retrain the original neural network setup with these revised input parameters. However, the performance of the D22 baseline is consistent with the performance metrics given in Dawson et al. (2022). Each time the network is trained produces slightly different results due to

the random split of the data into training, validation and testing and the random initialization of the CNN's weights and biases. To mitigate the effects of this randomness in the training process, we train each configuration ten times and calculate the mean and standard deviation of those ten runs (Table 1). For the following analysis we pick the neural network that yields the best test accuracy out of these 10 runs for each configuration.





| | Validation acc | Test acc | Lead user acc | Lead prod acc | TF user acc | TF prod acc | Ice user acc | Ice prod acc |
|---|---|---|---|---|---|---|---|---|
| D22 | 79% ± 4% | 77% ± 5% | 82% ± 10% | 81% ± 12% | - | - | 95% ± 3% | 95% ± 4% |
| More data | 83% ± 1% | 82% ± 2% | 84% ± 6% | 83% ± 6% | 82% ± 4% | 68% ± 11% | **97% ± 1%** | **97% ± 2%** |
| **This study** | **87% ± 1%** | **84% ± 2%** | **88% ± 5%** | **85% ± 6%** | **83% ± 6%** | **73% ± 9%** | **97% ± 1%** | **97% ± 1%** |

**Table 1.** Accuracy (acc) of different configurations on the validation data set, test data set and specifically for the lead class, the thinned floe (TF) class and all sea ice classes merged (thinned floe, good floes and noisy floes combined). The configuration proposed by Dawson et al. (2022) was rerun using slightly different input files and parameters as a baseline (D22, first row). We then increased the training data set ('more data', second row) and also tested the new CNN architecture ('this study', third row) to separate the impact of these improvements. For the individual classes and the combined ice class, we list both user and producer (prod) accuracies. The best agreement is highlighted in bold.

We observe an improvement for all metrics when the training data is extended and a further boost in accuracy when the novel
CNN architecture and training procedure are employed. The validation data is not directly used during the training process, but it was used to decide on the best architecture and determines when the training process is stopped. Therefore, it is not entirely independent. A big difference between validation and test accuracy would indicate that the CNN is overfitting; however, we do not observe this here with any model configuration. Both the validation and the test accuracy are increased by 4 and 5% respectively due to the increase in training data and by an additional 4 and 2% respectively due to the novel architecture
compared to the D22 baseline. Correctly identifying leads is most crucial for freeboard calculation. If ice is misclassified as a lead, this introduces a negative bias in the resulting freeboard. A higher lead user accuracy indicates that more of the samples classified as leads are actually leads. On the other hand, if leads are missed, this reduces the number of freeboard estimates. In this case, a higher lead producer accuracy indicates that more of the true leads are classified as such. For both metrics of the lead class, we observe an increase in accuracy by 2% when adding more data and by an additional 4 and 2% respectively
by also improving the CNN. This yields a lead user accuracy of 88% and a lead producer accuracy of 85%. The additional thinned floe class reaches a similar user accuracy of 82 and 83% depending on the CNN, but the producer accuracy improves from 68% to 73% thanks to the novel CNN architecture and training. As we have the least training samples for this class, the accuracies are slightly lower than for the other classes, but still encouraging. Finally, we present the combined sea ice accuracy, since confusing good floes with noisy floes or thinned floes has no impact on the freeboard calculation where the ice classes are
grouped together. The combined ice accuracy is highest for all configurations (as we have many more floe than lead samples), but still improves slightly from 95% in the original setup to 97% when the training database is extended. It is also striking how much the extension of the training database stabilizes the results (i.e. much lower variance in accuracy across runs). For the test accuracy, the standard deviation across the ten runs is reduced from 5% to 2% by extending the training database, and the standard deviations of the lead accuracies (user and producer) are reduced from 10 and 12% to 6% when more data is added.
The novel CNN architecture keeps the standard deviations around these same much lower values. This is an important result



because the network must be retrained every time the input data changes (e.g., for an updated CryoSat-2 baseline, or retracker implementation) and our results here suggest this can be done without re-evaluating the network architecture each time.

For the summer months, we can make a direct comparison to previous work by retraining the CNN developed by Dawson et al. (2022) with the same updated input parameters and training data and assessing it on the same test data. For other studies,
developed for the winter season, comparisons are less objective, since the test data is different and classification is generally an easier challenge in winter, before the melt onset. Dettmering et al. (2018) compared four different approaches for lead classification: Röhrs et al. (2012); Ricker et al. (2014); Müller et al. (2017); Passaro et al. (2018) and validated them with near-coincident optical imagery from airborne underflights. They tuned the parameters of each method with 50% of the data and report overall accuracies of around 97% for the remaining 50% of the data. The lead user accuracy (number of correctly
classified leads/number of classified leads) is found to vary between 21 and 39 % for the different approaches, while the lead producer accuracy (number of correctly classified leads/number of true leads), is only 15-23% for all approaches. Lee et al. (2016) state user accuracies between 48-95% and producer accuracies of 86-98% for two thresholding techniques (Laxon et al., 2013; Rose, 2013) and their own methods. In a later study (Lee et al., 2018), they then report user accuracies of around 83-91% and producer accuracies of around 88-95% for the same techniques and their new approach using a different test dataset. This
shows how much the reported accuracies vary across different test datasets. Nevertheless, when comparing to these studies, our novel CNN yields higher accuracies than reported by Dettmering et al. (2018), lower producer accuracies and similar or higher user accuracies than reported by Lee et al. (2018, 2016). Since we are calculating freeboard at the lead locations, it is especially important to reach a high user accuracy and given that classification is more challenging during the summer months, we regard these results as very encouraging.

## 4.2 Visual validation and interpretation

For a visual inspection and interpretation of the results, we classify one CryoSat-2 track from 16th June 2018 north of Greenland, crossing a near-coincident Sentinel-1 image with a time difference of only 5 minutes (Figure 6). Overall, the majority of points are classified as noisy floes, indicating variable along-track surface properties (Figure 3). We also observe sections of good floes, thinned floes and some lead points in between. Zooming in, we can confirm that the samples classified as leads
are indeed over areas that visually look like leads. There are no obvious examples of sea ice floes being erroneously classified as leads. However, it is apparent that the lead classification is rather conservative, regularly classifying samples as floes where there appears to be a lead in the coinciding image. This could be in cases where the CryoSat-2 footprint contains both water and ice, or when there are mixed conditions within the 11-point along-track window, meaning that the sample is classed as a noisy floe. Including these "floe" samples, that are actually from lead elevations, would act to bias the derived radar freeboards
slightly thin. However, the alternative possibility of erroneously classifying true floes as leads, for the sake of classifying more leads, leads to far more significantly underestimated freeboard. The thinned floe samples tend to be located in clusters outside the largest, most obvious floes, but there is no clear visual difference between the thinned floe and good floe samples in the Sentinel-1 image. During the generation of the training database, floe samples are separated into good, noisy and thinned floe classes based only on the along-track characteristics of the CryoSat-2 observations rather than any clear differences in the im-





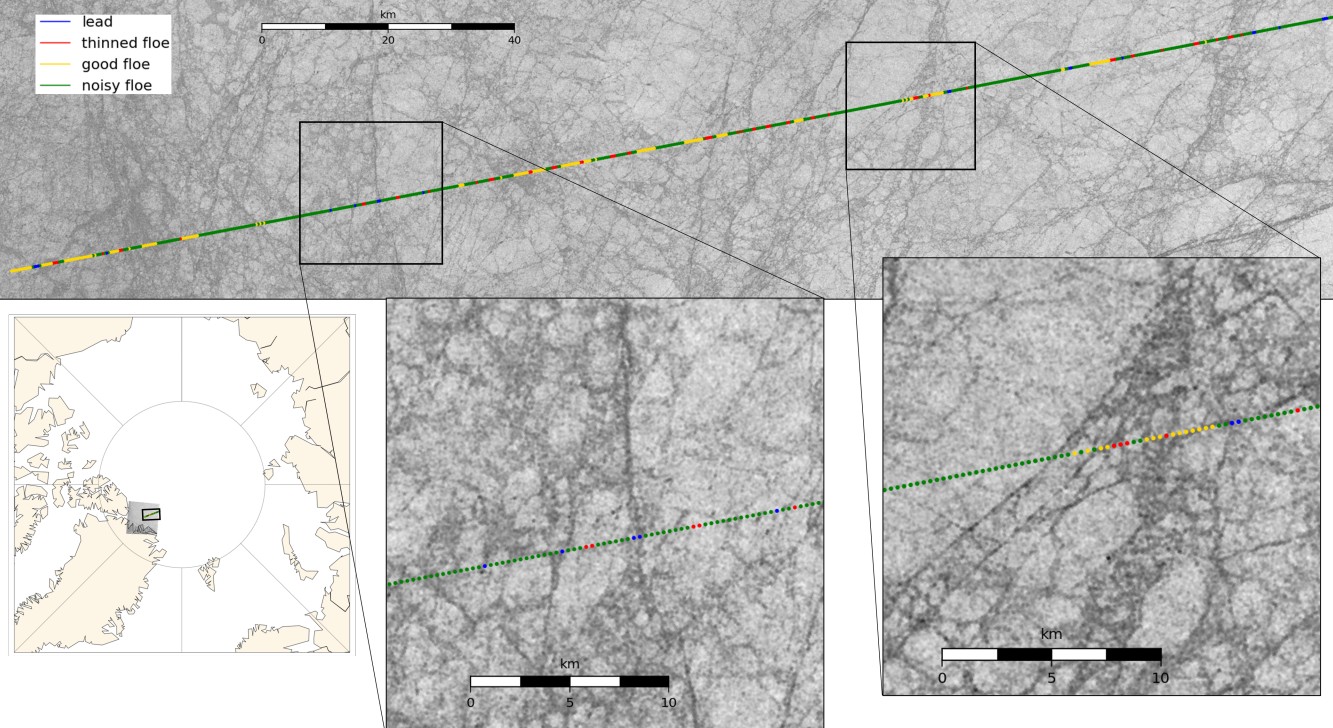

**Figure 6.** CryoSat-2 track (CS_LTA__SIR_SAR_2__20180616T181452_20180616T181735_E001) overlain on a Sentinel-1 image (S1B_EW_GRDM_1SDH_20180616T182229_20180616T182333_011402_014F10_457B) North of Greenland.

age data. When calculating freeboard, all three floe classes are merged to calculate the ice elevation, within a window around each lead, meaning that a confusion amongst them does not affect the freeboard estimate.

Finally, we notice that only 3-4 consecutive samples get classified as thinned floe, which is due to the class definition, where a local change in the waveform parameters is required for detecting thinned floe samples (Figure 3). As a result, the CNN cannot detect larger areas of consistently thinned floes.

## 4.3 Spatio-temporal evolution of class occurrence

For a comprehensive analysis of the novel classifier and physical interpretation of the different classes, we analyse the geographical spread and temporal evolution of all class fractions in 2015 (Figure 7). For leads, we observe that the lead fraction is lowest in July and August and around the margins. It is highest in the Laptev Sea in May and also rapidly increases from August to September across the whole Arctic except the margins. The lack of leads around the margins is related to the class definition (Figure 3), where leads are only detected when they are surrounded by higher elevation floes within the 3 km window. Leads that are too wide or adjacent to thin ice might be missed. This pattern is in contrast to winter classifiers, which commonly find most leads around the margins (Lee et al., 2018; Reiser et al., 2020; Ricker et al., 2016; Wang et al., 2016). Also




the lower lead fractions in the central summer months are related to the classifier design, where leads require a change in the waveform parameters alongtrack, whereas the returns from heavily melt pond covered sea ice are consistently dominated by pond reflections with little variation alongtrack. Our approach is designed to classify leads conservatively, to limit the number of lead commission errors that bias the freeboards thin, however this means that lead fractions in mid-summer are much lower than expected from sea ice concentration observations. The highest lead fractions observed in May and June are likely related to the sea ice opening up in these areas through divergence, as the thinnest ice from spring melts rapidly.

For the fraction of thinned floes, we observe clusters of higher thinned floe fractions, which evolve over the summer, but they are typically located outside of the Central Arctic and closer to the margins. In May 2015 most thinned floes are found East of Svalbard and Novaya Zemlya. By June, the areas of higher thinned floe fractions have expanded and cover large parts of the Laptev Sea up to the North Pole and in the Baffin Bay. In July the Beaufort Sea exhibits larger fractions of thinned floes, which intensifies in August. In September 2015 we still observe higher fractions of thinned floes in the remaining sea ice areas of the Beaufort Sea, but also around the other margins. These patterns support that the thinned floe class is related to physical properties of the sea ice and most often found in areas that are thinning, since it is clear from the maps that grid cells with high thinned floe fractions often become ice-free or lower concentration in the following month.

Good floes and noisy floes are generally much more abundant than leads and thinned floes (note different colorbar limits in Figure 7). The patterns of the good floe and noisy floe classes are very much reversed and while good floes are mainly found in July and August, noisy floes are most abundant in regions of thicker ice in May. Overall, this analysis suggests that as melt progresses, the waveform signature of floes often starts of as noisy floe, meaning there is variation in the waveform parameters along the track. With the onset of melt, the thinned floe fraction increases - meaning that the variation alongtrack is not arbitrary, but thinner floes can be identified. Finally, most floes exhibit waveform parameters that we classify as good floes and that exhibit little variation along the track as melt ponds dominate almost all return signals. September then yields the most diverse class distribution with comparably many leads, thinned floes and an about equal number of good and noisy floes. This change could indicate the drainage or refreezing of melt ponds and accumulation of snow, re-introducing more varied and less specular waveform responses again.

## 4.4 Resulting sea ice thickness

Next, we test the impact of the improved classification on the resulting SIT to verify that the novel approach works, when integrated into the whole sea ice processing chain. We use the best performing (highest test accuracy) CNN for each setup (see Section 4.1) and calculate SIT using the three different classification results (Figures 8, A1, A2).

The overall patterns and magnitudes are very similar for all three classification setups and seem realistic overall. The transition between the winter processing chain and the summer processing chain, which use different classifiers, different retrackers and calculate freeboard at the floe or lead locations respectively, appears smooth with persistent anomalies and comparable magnitudes (see Figure 8 for the full year 2015 and Figure A3 for the spring and autumn transitions of 2015-2019). In 2015 the transition from April to May looks slightly less smooth, with a higher increase in SIT north of the Canadian Arctic Archipelago





**Figure 7.** Geographical spread of class fractions (columns) between different months (rows) for 2015



when the original CNN (D22) is used (Figure A1) compared to the novel architecture (Figure 8), however, the differences are small.

When a direct comparison is made between the different versions, we observe that the novel CNN generally increases SIT across all months and regions except a slight decrease North of the Canadian Arctic Archipelago in May and mixed signals around the margins in July and August. On average, the SIT obtained from freeboards with the original architecture and more data is $11 \pm 4$ cm thicker than the SIT obtained from D22 freeboards, whereas the SIT obtained from the new CNN and more data is $4 \pm 6$ cm thicker than D22. The highest increase is observed in the first half of September, but rarely exceeds 50 cm difference (Figure A4).

The most striking difference between the three setups is the closure of data gaps when more data and the new CNN are used. When the original D22 setup is used, no valid freeboard estimates are produced for some sea ice covered areas in the Chukchi Sea from mid June (grey areas in Figure A1). The data gaps are largest for the D22 setup in July and persist until end of September. The 'more data' setup reduces the areas where data are lacking, but data gaps are still apparent - especially in the second half of July (Figure A2). Using the novel CNN closes most of the remaining data gaps (Figure 8). The differences indicate that more leads are detected by the CNN when the training database is expanded and even more when the novel CNN architecture is used. Another aspect could be, that less samples are filtered out because of high errors or unrealistic values. In any case, this is a desirable advance - especially when it affects the marginal ice zone and thinner ice areas, as these are of highest interest for marine activities, forecasting, and air/sea/ice interactions including biological productivity.

## 4.5 Validation and comparison of resulting sea ice freeboard, thickness and volume

In order to quantify the impact of the improved classification on the resulting freeboard, SIT, draft and sea ice volume, we compare estimates from each setup to various validation datasets as well as PIOMAS (Table 2).

When the different classification approaches are used to calculate freeboard, draft, SIT and SIV, the impact of the classification on the agreement with in situ and modelling data is minor. This is because various additional uncertainties are introduced when freeboard is calculated (e.g. from retracking, averaging floe heights, interpolating to in situ data, not fully compensating for the EM bias and the uncertain penetration depth of the radar signal in an evolving snow pack). External datasets are required for the bias correction and the snow speed/penetration correction and come with their own uncertainties. When comparing SIT or draft, ice density and potential melt pond loading further add to the uncertainty budget. Finally, when SIV is compared, gap filling and interpolation as well as the additional external sea ice concentration estimates further add to the uncertainty budget.

Despite all these shortcomings, we observe a minor, but consistent improvement in the comparison to five out of six validation and comparison datasets, when the training database is expanded. When the novel architecture is introduced, the metrics either stay the same or get minimally worse compared to 'More Data', but are still better or the same as D22. The only exception is the comparison to the Barents Sea moorings, where the RMSE and Pearson correlation coefficient are marginally best for the original setup (D22, Table 2). On the other hand, here the novel architecture yields the smallest bias overall (0.00 m). Across all validation and comparison data sets, the differences observed are very small and likely smaller than the impact of





**Figure 8.** Sea ice thickness [m] across the Arctic for 2015 at twice monthly intervals using the method described in Landy et al. (2022) and the novel CNN architecture for the summer (May-September)



|  |  | OIB fb (2016-2017) | AEM SIT (2016-2018) | BGEP draft (2015-2019) | Laptev draft (2015-2016) | Barents draft (2015-2019) | PIOMAS SIV (2015-2019) |
|---|---|---|---|---|---|---|---|
| D22 | Bias | -0.06 m | -1.20 m | -0.32 m | -0.28 m | -0.04 m | -3330 km$^3$ |
|  | RMSE | 0.08 m | 1.54 m | 0.51 m | 0.62 m | **0.48 m** | 3670 km$^3$ |
|  | R | 0.88 | 0.30 | 0.82 | 0.36 | **0.39** | 0.97 |
|  | n | 32 | 189 | 150 | 37 | 68 | 50 |
| More data | Bias | **-0.05 m** | **-1.11 m** | **-0.27 m** | **-0.21 m** | 0.01 m | **-2330 km$^3$** |
|  | RMSE | **0.06 m** | **1.45 m** | **0.48 m** | **0.56 m** | 0.50 m | **2880 km$^3$** |
|  | R | 0.88 | **0.31** | **0.83** | **0.43** | 0.38 | 0.97 |
|  | n | **35** | **194** | 150 | 37 | 68 | 50 |
| This study | Bias | -0.06 m | -1.14 m | -0.29 m | -0.25 m | **0.00 m** | -2690 km$^3$ |
|  | RMSE | 0.08 m | 1.49 m | 0.49 m | 0.60 m | 0.49 m | 3180 km$^3$ |
|  | R | 0.88 | **0.31** | **0.83** | 0.39 | 0.38 | 0.97 |
|  | n | **35** | **194** | 150 | 37 | 68 | 50 |

**Table 2.** Comparison of CryoSat-2 derived sea ice freeboard (fb), sea ice thickness (SIT), sea ice volume (SIV) and draft to validation and model data (Figure 2 and Section 2.3) using the three different classification setups. The configuration proposed by Dawson et al. (2022) was rerun using slightly different input files and parameters as a baseline (D22, first row). We then increased the training data set ('more data', second row) and also tested the new CNN architecture ('this study', third row) to separate the impact of these improvements. For each comparison the bias, the root mean squared error (RMSE), the Pearson correlation coefficient (R) and the number of samples (n) are given. The best agreement is highlighted in bold.

other uncertainties in the processing chain, which mask the improvement in classification between the 'more data' setup and
425 the novel CNN setup ('this study'), which was seen from the test data (Table 1).

As discussed by Dawson et al. (2022) and Landy et al. (2022), the seasonal cycle is well captured with all three setups (R=0.97 compared to PIOMAS and R=0.82-0.83 compared to the BGEP moorings). Figure 9 shows the timeseries of ice draft measured by the BGEP moorings and the CryoSat-2 based ice draft estimates using the three different classification setups. Also here, we observe generally good agreement between the moorings and the CryoSat-2 estimates and a smooth transition
between the winter (green) and summer (blue) estimates. In some years the timing of the melting agrees very well with the in situ observations (e.g. 2018, 2019), whereas in other years the CryoSat-2 based estimates drop before the ULS based estimates (e.g. 2015). This could also be a result of the large-scale averaging, or remaining biases in the accounting of snow and meltwater loading of the ice. In some years there are significant biases against the BGEP ULS data, for instance in 2016 across the winter and summer at moorings A and D across all three versions. This points to biases in either/both the auxiliary snow loading
information or the assumption of 90-100% radar penetration through snow. Concerning the three different classification setups, the differences are very small and no best setup can be identified visually.



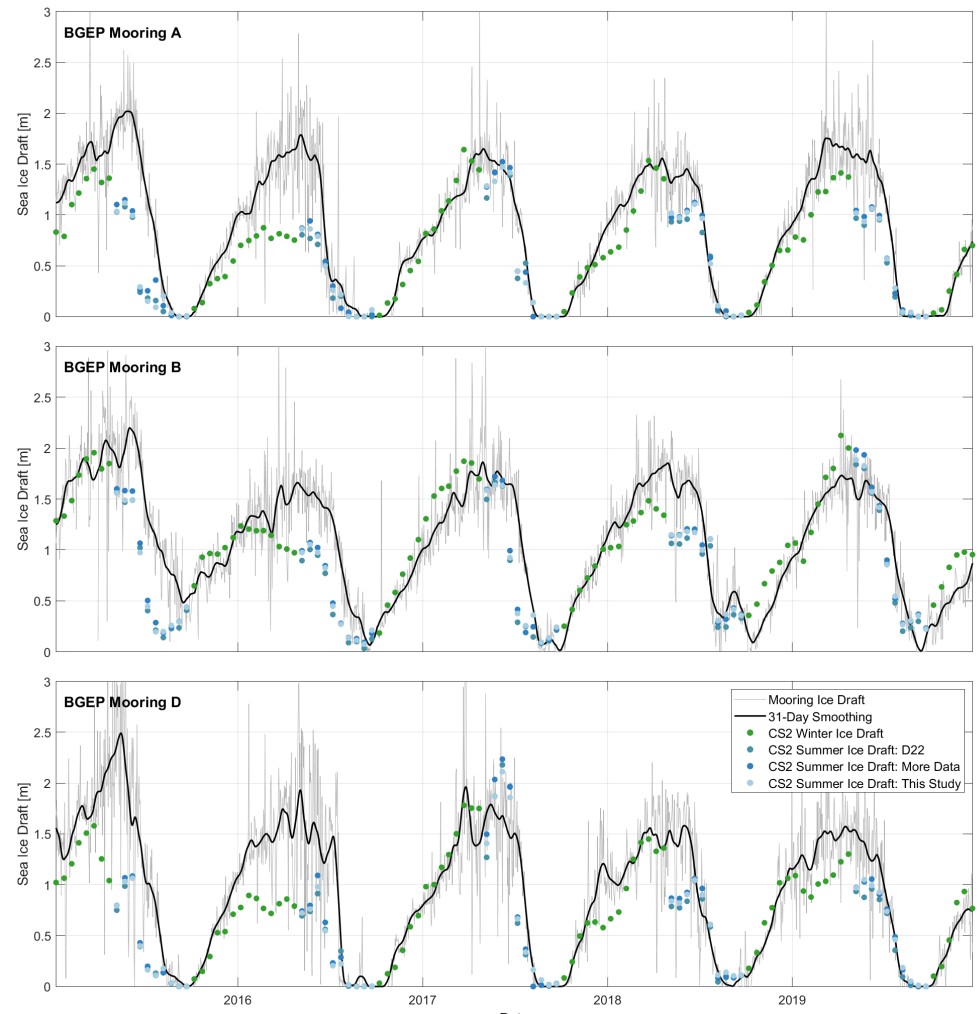

**Figure 9.** Time series of sea ice draft measured by the BGEP moorings and calculated from CryoSat-2 freeboard observations around the mooring positions. Winter ice draft (green dots) is calculated as described in Landy et al. (2022) and for summer ice draft we plot three versions using the different neural network configurations (blue dots).

From these comparisons we conclude that the classification has reached a satisfactory accuracy (comparable to winter) and that other factors like retracking, snow depth and the EM bias correction have a larger potential to make significant changes to the resulting SIT. The novel classification has, however, further improved our confidence in the lead classification, increased
the number of resulting freeboard estimates - especially in the marginal ice zone - and the additional thinned floe class opens up new opportunities to exploit it in future research.





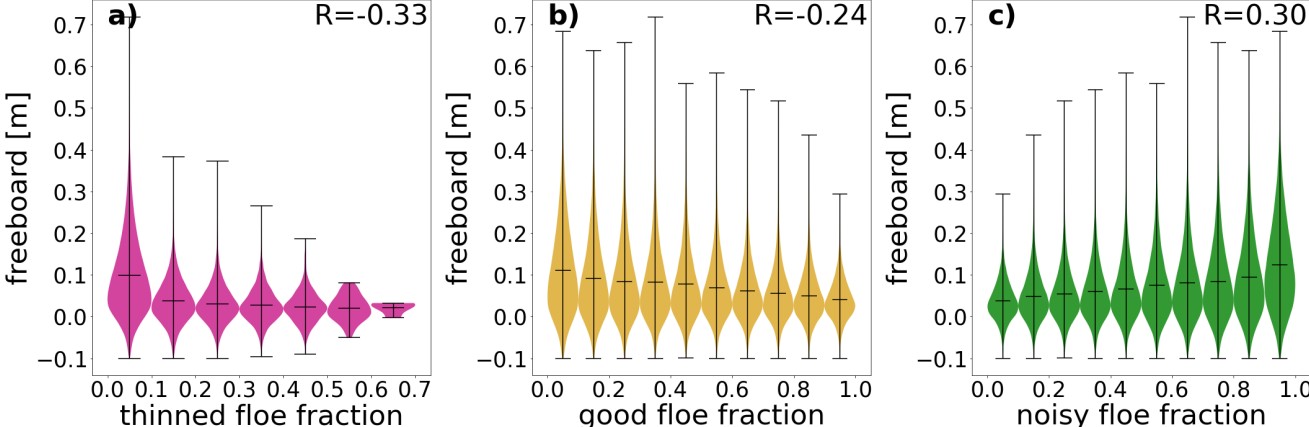

**Figure 10.** Correlation between the fraction of thinned floes (a), good floes (b) and noisy floes (c) with the freeboard height distribution for April - October 2018.

### 4.6 Thinned floe validation

To validate that the novel thinned floe class actually corresponds to thinner sea ice, we plot the fraction of thinned floes, good floes and noisy floes used per freeboard estimate (i.e. within 7 km around each lead) against the freeboard height distribution for April - October 2018 (Figure 10). We observe a wide spread of freeboard estimates (between -0.10 and 0.72 m), when the fraction of thinned floes is below 10%. This spread gradually decreases as a larger proportion of the floes are thinned floes. For the highest observed thinned floe fractions of between 60-70%, freeboard only varies between 0.00 and 0.03 m. Overall, the Pearson correlation coefficient between the thinned floe fraction and freeboard is -0.33, indicating that what the CNN identifies as thinned floes is indeed thinner than average and agrees with how the class was designed. We point out, however, that these thinned floes are only detected when they are *thinner* than their surrounding ice (with an obvious along-track anomaly, Fig. 3), so this class will not capture all of the absolute thinnest ice. A portion of the absolute thinnest ice will be classified as noisy or good floe, if there are larger areas of consistently thin ice, most likely as good floe since we expect low along-track variability for ice with consistently low freeboard.

The same analysis applied to the fraction of good floes and noisy floes also reveals that the good floes exhibit a negative correlation (R = -0.24) with freeboard height, albeit with a weaker relationship than for thinned floes (Fig. 10). This indicates that good floes tend to represent relatively thinner freeboards than average. In contrast, a higher noisy floe fraction is positively correlated (R = 0.30) with higher freeboards. The difference between the good and noisy floe classes is mainly related to the degree of melting/melt pond coverage and hence has a seasonal evolution. The more melt ponds are present, the more they dominate the return signal and equalize the waveform parameters alongtrack. Therefore, good floe samples mainly stem from the peak summer months, when the sea ice is thinnest (see Section 4.3, Figure 7). The patterns for good and noisy floes also mirror each other in Figure 10, as they do in Figure 7.



## 5 Conclusions

In this study, we improved the classification of CryoSat-2 measurements into leads and floes for the summer months. We built on the approach by Dawson et al. (2022), diversified and increased the amount of training data by more than fivefold,
advanced the CNN architecture and training procedure, and introduced an additional class for thinned floes. We showed that both the expansion of the training data and the novel CNN boost the accuracy of the classification across all metrics. The lead user accuracy, which is the key parameter for freeboard calculation, increased from $82 \pm 10$ % to $84 \pm 6$ % thanks to the multiplication of the training samples and reached $88 \pm 5$ % with the novel CNN employed. Our analysis also showed that the algorithm is still conservative, i.e. the detected leads mostly correspond to real leads. On the other hand, true leads may be
missed when the footprint contains mixed surface types, around the margins and in the central summer months. The latter two are related to the class definition, which expects a clear change in elevation and waveform parameters along the track in order to find a lead.

When using the novel classification method to calculate sea ice freeboard, thickness, draft and volume during the summer months, we found realistic spatial and temporal patterns and consistency at the transitions between winter and summer pro-
475 cessing in spring and autumn. Nevertheless, the impact of the improved classification on the resulting SIT is minor and when comparing to various in situ and model datasets, we find only marginally improved agreements. The main advancement for SIT is the extended coverage, as more leads are detected with the novel CNN - especially in the marginal ice zone. In order to further reduce the bias over rough multi-year ice and therefore improve the agreement with airborne data from this region, advancements to other parameters in the processing chain need to be developed. Specifically, improvements made to the re-
480 tracking algorithm, estimates of snow depth, density and penetration, and the EM bias correction all have potentially bigger impact on the overall SIT.

Finally, we demonstrated that the novel thinned floe class is correlated with thinner ice than average and that the thinned floe fraction exhibits patterns related to melting. However, due to the along-track class definition, the CNN only picks up floes that are thinner than their surrounding rather than the absolute thinnest portion of the sea ice. The class was introduced to
485 aid classification as it exhibits unique changes in the along-track waveform parameters. We anticipate that it might also prove useful for future research on summer sea ice freeboard and SIT, though.

*Data availability.* The updated and expanded training database is available from 10.5281/zenodo.15645704



**Figure A1.** Sea ice thickness across the Arctic for 2015 at twice monthly intervals using the method described in Landy et al. (2022) and the original CNN architecture from Dawson et al. (2022) for the summer (May-September)



**Figure A2.** Sea ice thickness [m] across the Arctic for 2015 at twice monthly intervals using the method described in Landy et al. (2022) and the original CNN architecture from Dawson et al. (2022) using more training data for the summer (May-September)



**Figure A3.** Sea ice thickness [m] across the Arctic at the transition between winter and summer processing during spring (April/May) and autumn (September/October) for five years (2015-2019) using the novel CNN



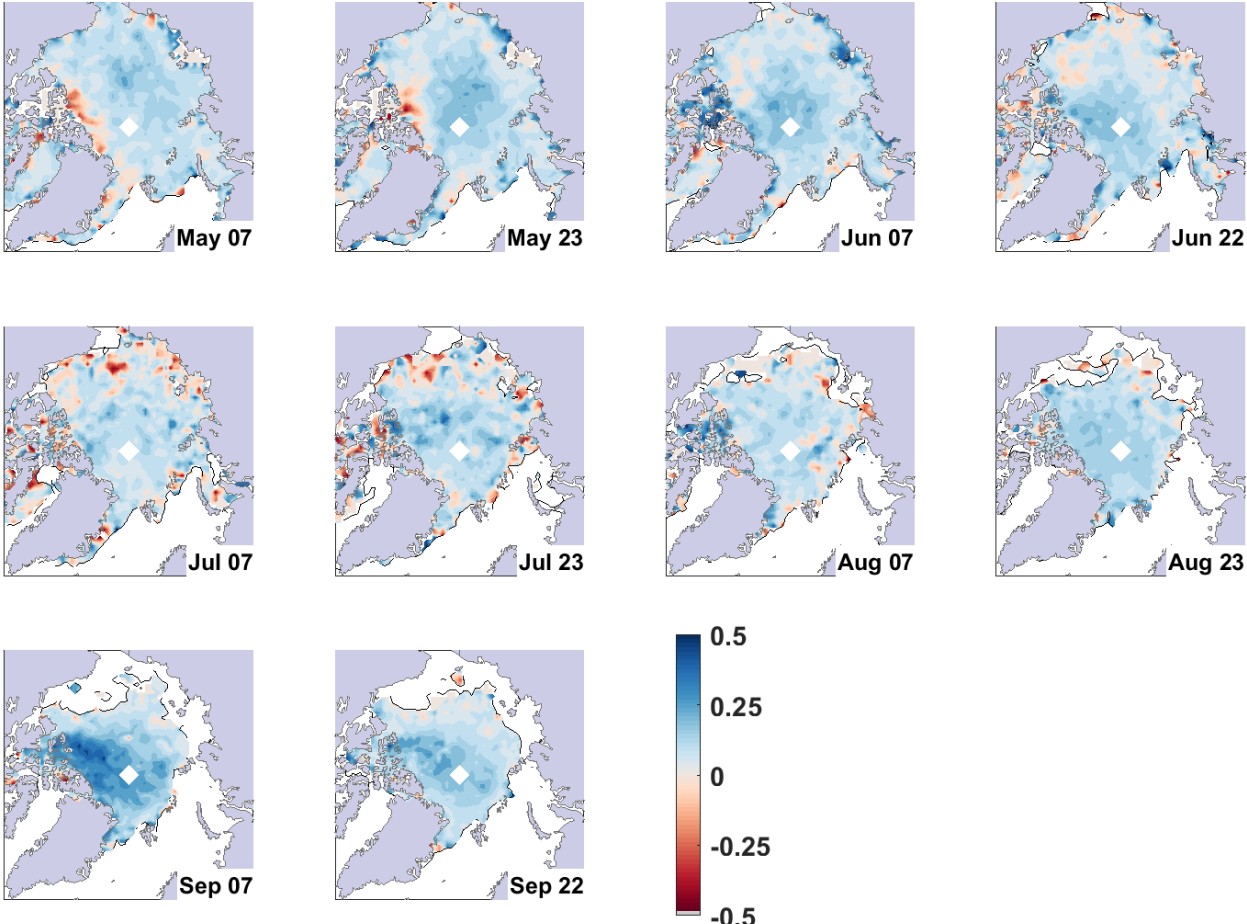

**Figure A4.** Difference in SIT [m] when the original (D22) setup or the novel CNN (this study) are used for classification. Each map represents the average over two weeks across 5 years (2015-2019) labeled with the middle date of each 2 week period.

*Author contributions.* All authors contributed to the design of the study. JL and GD extended the training database, ABF designed the novel CNN and trained the three network setups. ABF, JL and GD developed code for freeboard and SIT calculation and ABF and JL generated the figures. ABF, JL and RR discussed the result and wrote the paper.

*Competing interests.* The authors have no competing interests.



*Acknowledgements.* ABF and JL were funded by the Research Council of Norway (NFR) through the InterAAC project (328957) and by ESA through the CryoTEMPO project (Contract AO/1-10244/2-/I-NS). ABF, JL and RR also received funding by ESA's CLE2VER project (Contract AO/1-11448/22/I-AG). RR was supported by ESA CCI Sea Ice (CCN-2 to Contract 4000126449/19/I-NB -Sea_Ice_cci) and RR,
JL and ABF by the Fram Centre project "Sustainable Development of the Arctic Ocean" (SUDARCO) (project_ID: 2551323).



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
