# Peer review of "Enhanced neural network classification for Arctic summer sea ice"

_EGUsphere, 2025_

## Author Comment (AC1)

Dear Sanggyun Lee,

Thank you very much for taking the time to review our manuscript and your constructive and supportive comments. They certainly helped us improve the quality and clarity of the paper. Please see our responses in blue below.

This study follows Dawson et al. (2022) and Landy et al. (2022), extending their work by adding more training samples and improving the CNN architecture. While the contributions are clear, there remain parts of the Methods section that require further elaboration to improve readability and reproducibility. I therefore recommend publication after a major revision.

**General comments:**

**P4, L97**: Since the addition of training samples is a key contribution of this study, I recommend providing a table summarizing the newly added samples, including their geographic region, month, and satellite source.

We agree that this is a key contribution and that all the listed information is relevant. You can find the novel training dataset on zenodo (as csv file) and linked to the paper in the assets. This table contains a binary flag indicating whether the sample was newly added or not, the coordinates, time, class label, names of the corresponding image and CryoSat-2 files and all input variables within the 11-point windows.

**P4, L117**: Could the authors clarify why a 7 km window was chosen?

As explained in the text, we average the airborne observations along 7-km sections to mirror the length scale used in obtaining a single freeboard estimate from CryoSat-2 for our summer CryoSat-2 processor (L117-118, L243). In the summer CryoSat-2 processor this window size was chosen, because it "[…] included a sufficient number of floe elevation samples to obtain realistic freeboard measurements even when there was high local variability in elevation." (Dawson et al. 2022, end of page 5).

**P8, L172**: Apparently, the lead and ice samples were manually extracted using visual inspection. However, as noted in Section 4.2, distinguishing between thinned floes and good floes is not straightforward in Sentinel-1 imagery. Could the authors comment on the potential impact of human error during the manual extraction of training samples, and how such uncertainty was mitigated?

We added the following explanations at the end of the section in the paper. "There is the possibility for human error in the manual extraction of training samples; however, several steps were taken to mitigate this. Most care was taken in the identification of leads, since the other ice floe classes are eventually combined in the freeboard processing. Leads were identified where there was any evidence for variation in CryoSat-2 SSH around the sample, and where the lead could be clearly visualized in the coincident image, from both a contrast in intensity and – importantly also – by the shape

of the feature. There is more chance for human error in the separation between ice floe classes, but these are radar classes: for all floes the sample was clearly an ice floe in the coincident satellite image, but it was only based on the patterns in the along-track CryoSat-2 parameters that they were separated into thinned floe, good floe and noisy floe classes."

**P8, L175**: If I understand correctly, Istomina et al. (2016) does not explicitly demonstrate that melt ponds and leads can be spectrally distinguished. It may be helpful for the authors to clarify how this reference supports their statement on spectral separation.

We replaced the reference with Istomina et al. 2025, where Figure 8 shows the albedo per wavelength, making it clear that melt ponds are generally more reflective, but especially in the blue wavelengths.

**L8, 187**: Since distinguishing leads from melt ponds is the most critical challenge in summer, could the authors clarify whether melt ponds are assumed to be entirely included in the noisy floe class? In addition, are refreezing ponds in August and September also included in this class? From a sea surface height (SSH) perspective, it may be difficult to separate thinned floes from refreezing ponds. It would be helpful if the authors could explain how refreezing ponds are treated in their classification.

Thank you for this comment and pointing out this confusion. We now made it more explicit throughout the paper, that only the differentiation between leads and floes is physically meaningful. All ice classes (thinned floe, good floe, noisy floe) should be regarded as radar detectable classes that might not have a clear physical meaning or where the mapping to their corresponding ice class is yet to be understood. To explicitly answer your question: Ponds and refrozen ponds could in principle be included in any of the floe classes, as they cannot be reliably detected in the CryoSat-2 or image data and therefore do not form a separate class in the training dataset. We separated leads and floes from the image data and assigned the three floe classes based on their unique signatures in the CryoSat-2 data. After classification, all floe classes are merged to calculate freeboard.

**Figure 3**: It would strengthen the manuscript if the authors could provide quantitative evidence that the addition of the thinned floe class improves the classification performance.

We did not detect a significant quantitative difference between using three or four classes (i.e. adding the thinned floe class) but decided to treat the thinned floe samples as a separate class, since the class reveals unique patterns in the along-track waveform parameters and even if we do not yet fully understand it physically, it opens up novel research avenues. Anyone using the CNN is free to experiment with the new class, to simply treat it as one of three ice classes or to discard it.

**Figure 6**: Since surfaces with diverse geophysical conditions also occur in July, August, and September, it would be helpful to include additional examples of CryoSat-2 tracks with coincident imagery from these months. This would further support the robustness of the classification.

Thank you for this suggestion. We added two additional examples from July and September.

**Minor comments:**

**P1, L16**: One reference is missing the publication year. Please correct this.

Done

**Figure 1**: The caption of Figure 1 is unclear as currently written. Please rephrase it to improve clarity.

Done

**P8, L163-164**: The text refers the reader to Dawson et al. (2022), but without further explanation it may be difficult to understand Figure 3 and the use of the 11-point window. A short description would be helpful.

Good point. We added a short explanation here: "For each sample, the anomalies of five parameters are extracted along an eleven-point window centered around the labeled sample to capture along-track anomalies."

**P10, L209**: Please specify whether pooling refers to max pooling or mean pooling.

We mean max pooling (l.207) and explicitly rephrased pooling to max pooling everywhere in the text now.

**P10, L221**: Did the authors also test the Adam optimizer? If so, was there any improvement in performance compared to RMSProp?

Yes, we also tried the Adam optimizer and found no improvement, so we stuck to RMSProp, which was also used by Dawson et al. (2022).

**Table 1 and 2**: Please place the captions above the tables rather than below.

Done

**Table 1**: For the ice user/producer accuracies, does this metric include both good and noisy floes together? Please clarify.

Yes, we clarified this in the text.

---

## Author Comment (AC2)

Thank you very much for taking the time to review our manuscript and your constructive and supportive comments. They certainly helped us improve the quality and clarity of the paper. Please see our responses in blue below.

**General Comments**

In this manuscript, the authors build on the summer CryoSat-2 lead/floe classification convolution neural network (CNN) of Dawson et al. (2022) by expanding and improving the training dataset and introducing a new thin-ice classification. They then evaluate Arctic summer sea ice thickness and volume using this improved lead/floe classification. This is an important challenge for the sea ice community as melt ponds on summer sea ice complicate classification of the sea surface in radar altimetry, and historically, sea ice thickness estimates excluded summer for this reason. The authors find an increase in accuracy when using the new dataset, which is promising for improving current summer sea ice thickness retrievals.

Overall, this manuscript is well-written and relevant to the wider science community, where desire for year-round Arctic sea ice observations is high. However, I do have some general concerns, particularly regarding the thin ice classification.

1.  My biggest concern arises from Fig 3., where different waveform parameters are plotted for leads, good floes, noisy floes, and the new thinned floes classification. At present, I am not convinced by the characteristics used to determine the thinned floe classification. I would not expect a decrease in pulse peakiness (PP) over thinned floes, but rather an increase compared to good floes. We tend to see a reduction in peak power over thick floes and thin floes are likely to return a more specular waveform and thus have a higher PP (e.g., Rinne and Similä, 2016; Laxon, 1994). Zygmuntowksa et al. (2013) find that PP values from thin ice and leads are similar. I think further evaluation and clarification of this new class is required.

    Thank you for your comment and pointing out that the class name might be confusing. We now added an explicit comment in the paper just after introducing the thinned floe class that this class is not to be confused with thin ice forming in winter, but that it rather corresponds to thinned, rotten ice that is about to decay (see Figures 7 and 10). Furthermore, we made it more explicit throughout the paper, that only the differentiation between leads and floes is physically meaningful. All ice classes (thinned floe, good floe, noisy floe) should be regarded as radar detectable classes that might not have a clear physical meaning or where the mapping to their corresponding ice class is yet to be understood.

2.  The lack of leads in the central Arctic in July and August (Fig 7.) is noted as likely an artefact caused by the conservative treatment of leads in the classifier design,

which I understand. However, this does make me doubt its performance in the peak of summer, as there are areas of almost 0% leads during these months, but almost 100% good floes. Does this mean sea ice freeboard/thickness cannot be calculated at all in these grid cells? What happens if the classifier is relaxed for leads? Have you assessed against imagery in these cases, for example? What are the classifier performance statistics during these months specifically?

As you can see from Figure 8, we have enough lead samples to estimate freeboard in most of the grid cells. There are only a few small gaps in July that result from the lack of leads, but in all other cases a few leads are still found. We also calculated the performance statistics per month and added a comparison to a Landsat 8 image in mid July (see your later questions for more details).

3. I think an opportunity has been missed in Fig. 6 to provide a more in-depth visual comparison to imagery. You have included one SAR image and a qualitative assessment in Section 4.2., but ~3 examples would be useful here so we can get a better understanding of how the new classification performs particularly if the ice is more complex or during July/August given the poorer lead detection in those months, rather than fairly consolidated ice as it is in this example.

Thank you for this suggestion. We added two additional images from July and September, where July is a comparison to a Landsat image, exhibiting melt ponds, and the September Sentinel-1 image exhibits a larger network of leads.

**Specific Comments**

Figures – I note specific cases below, but generally the text and symbols on the figures are too small, and at times the labels are overlapping/not visible. Please fix.

Done

Line 90 – Is this limit based on the CryoSat-2 footprint?

Yes, we explained this in more detail in the paper now: "All images were retained when the time difference was <15 minutes. Between 15- and 60-minute time difference, we retained all images when ice drift speed was low (<0.1 m/s, corresponding to the along-track footprint size of 360 m displacement in 60 minutes). For higher ice drift speeds, we checked the visual alignment between features in the along-track CryoSat-2 data and removed images if they were apparently misaligned, to ensure we only use unambiguous samples for training. All images were removed when the ice drift was >0.3 m/s, corresponding to 1080 m displacement in 60 minutes."

Fig. 1 – I'm not sure if there is a benefit to having the sea ice freeboard grid underneath the points, especially as it isn't referenced in the text or caption. I think it makes the points harder to see. In the legend boxes, please also make the point symbols larger.

We decided to leave the background sea ice freeboard for reference, but added a reference to it in the caption and made the symbols larger.

Lines 149-151 – These sentences would benefit from rephrasing; it states that there are a lack of summer snow depth datasets but then introduces the SnowModel-LG without context of how this modelled data can therefore be produced.

The sentence says there is a lack of OPERATIONAL summer snow depth datasets and that satellites struggle to measure snow depth during the melt season. We clarified that SnowModel-LG is also non-operational, but available until July 2021 and that it is based on reanalysis data in the next sentence now: "[...] SnowModel-LG, which is based on reanalysis data, and non-operational, but currently available until end of July 2021 from NSIDC [...]"

Lines 171-172 – What is meant by 'misaligned tracks'? How do you determine if a track is misaligned? I'm concerned this would mean only tracks which seem to agree with the underlying image are retained.

There were very few samples that were discarded for this reason. However, what was meant by this is that we only included unambiguous samples in the training database, and discarded tracks where the time difference or drift were too big to match the CryoSat signal to the image (see also the previous response to L90). We clarified this in the paper.

Fig. 3 – Some of the figure labels are overlapping, which I appreciate is hard given the number of boxes. Do all of the axes need labelling, especially as for each row they are the same?

Done, thanks for the suggestion to simplify the labels!

Table 1 – Is it possible to include monthly performance statistics for summer? I would be interested in how the lead classification performs in July/August based on my comment above.

We calculated the classification accuracies from Table 1 per month to assess whether the performance is degraded for July or August, but found consistent results:

| | samples | Test acc | # leads | Lead user | Lead prod |
|---|---|---|---|---|---|
| All months | 263 | 84 ± 2% | 44 ± 6 | 88 ± 5% | 85 ± 6% |
| May | **18 ± 4** | 88 ± 9 % | 5 ± 2 | 91 ± 17 % | 93 ± 9 % |
| June | 41 ± 6 | **81 ± 5 %** | 7 ± 2 | **79 ± 17 %** | 85 ± 14 % |

| | | | | | |
|---|---|---|---|---|---|
| July | 64 ± 10 | 88 ± 4 % | 4 ± 1 | 95 ± 10 % | **69 ± 16 %** |
| August | 54 ± 5 | 87 ± 7 % | 4 ± 2 | 93 ± 17 % | 79 ± 23 % |
| Sep | 86 ± 9 | 83 ± 4 % | 24 ± 6 | 86 ± 5 % | 93 ± 4 % |

We have a different amount of samples per month and some variation in it, depending on how the training, validation and test data are split up. Naturally, months with less samples exhibit more variability (e.g. May has a much higher spread in the overall test accuracy than September). This becomes even more obvious when we separate leads per months, as often only a few samples exist per month, making the results very dependent on which few leads were selected as test data. Due to the small amount of samples, the results can only give an indication, but overall we find good agreement between the individual months and the overall performance (within the spread). The lowest test accuracy and lead user accuracy is reached for June, but given the wide spread, these results are not abnormal. For the lead producer accuracy we find that July performs worst, confirming what we observe from the plots that our CNN is particularly conservative in the central summer months.

We also added this table to the appendix and included a few sentences on this when discussing the low lead fraction in central summer.

Lines 399 – 401 – Yes, there is a substantial improvement in coverage, well done!

Thanks

Section 4.6. – I think this section should come higher up in the text as it doesn't feel as relevant at the end of the sea ice thickness/volume validation.

Good point. We rearranged the sub-sections as suggested.

**Grammatical Comments**

Line 16 – Incomplete reference. This reference is also linked to a dataset, rather than a paper in the full reference list.

Well spotted - we fixed it

Line 59 – Reference needs brackets.

Done

Line 89 – Reference needs brackets.

Done

Line 109 – Reference needs brackets.

Done

Lines 115-119 – There is some repeated text here from the paragraph above.

Well spotted. We removed the duplications.

Line 142 – PIOMAS acronym has already been defined.

Done

References – There are several incomplete references/DOIs. Please check.

Fixed this

Dawson, G. et al. (2022). A 10-year record of Arctic summer sea ice freeboard from CryoSat-2. Remote Sensing of Environment. 268, 112744. doi.10.1016/j.rse.2021.112744

Rinne, E. and Similä, M. (2016). Utilisation of CryoSat-2 SAR altimeter in operational ice charting. The Cryosphere, 10, 121-131. doi:10.5194/tc-10-121-2016

Laxon, S. (1994). Sea ice altimeter processing scheme at the EODC. International Journal of Remote Sensing, 15:4, 915-924. doi.10.1080/014311694408954124

Zygmuntowska, M. et al. (2013). Waveform classification of airborne synthetic aperture radar altimeter over Arctic sea ice. The Cryosphere, 7, 1315-1324. doi:10.5194/tc-7-1315-2013.